# Relation between Photogrammetry and Spinal Mouse for Lumbopelvic Assessment in Adolescents with Thoracic Kyphosis

**DOI:** 10.3390/healthcare12070738

**Published:** 2024-03-28

**Authors:** Guido Belli, Luca Russo, Mario Mauro, Stefania Toselli, Pasqualino Maietta Latessa

**Affiliations:** 1Department of Sciences for Life Quality Studies, University of Bologna, 47921 Rimini, Italy; guido.belli@unibo.it (G.B.); stefania.toselli@unibo.it (S.T.); pasqualino.maietta@unibo.it (P.M.L.); 2Department of Human Sciences, IUL Telematic University, 50122 Florence, Italy; l.russo@iuline.it

**Keywords:** lumbosacral region, postural balance, photogrammetry, adolescents, kyphosis, regression models

## Abstract

The evaluation of the lumbopelvic region is a crucial point during postural assessment in childhood and adolescence. Photogrammetry (PG) and Spinal Mouse (SM) are two of the most debated tools to properly analyze postural alignment and avoid misleading data. This study aims to find out the best linear regression model that could relate the analytic measurements of the SM with one or more PG parameters in adolescents with kyphotic postures. Thirty-nine adolescents (female = 35.9%) with structural and non-structural kyphosis were analyzed (13.2 ± 1.8 years; 1.59 ± 0.12 m; 47.6 ± 11.8 kg) using the SM and PG on the sagittal plane in a standing and forward-bending position, allowing for the measurement of body vertical inclination, lumbar and pelvic alignment, trunk flexion, sacral inclination during bending, and hip position during bending. Lordosis lumbar angles (SM) were significantly (r = −0.379, r = −0.328) correlated with the SIPS-SIAS angle (PG) during upright standing, while in the bending position, the highest correlation appeared among the sacral–hip (SM) and the sacral tangent (ST_PG; r = −0.72) angles. The stepwise backward procedure was assessed to estimate the SM variability in the bending and standing positions. Only in the bending position did the linear regression model reach high goodness-of-fit values with two regressors (ST_PG $\eta^{2}=0.504$, BMI $\eta^{2}=0.252;$ *adjusted- R*^2^ =0.558, *p* < 0.001, CCC = 0.972, r = 0.763). Despite gold-standard methods reducing error evaluation, physicians and kinesiologists may consider photogrammetry as a good method for spinal curve prediction.

## 1. Introduction

The lumbopelvic region is a fundamental component of human body posture [1]. This area includes all bone and myofascial tissues that connect the lordotic spine and pelvis with the upper and lower parts of the body. Globally, the musculature involved is defined as the “lumbopelvic–hip complex (LPHC)” and plays a crucial role in (1) the transfer of energy between the proximal and distal part of the body along the kinetic chain, (2) enhancing trunk stability and muscle coordination, and (3) protecting the spine and balance joints load [2,3,4,5]. 

LPHC stability is defined as the ability to control the position and motion of the trunk over the pelvis and leg, to maintain postural control, and guarantee proper body alignment [2,6]. Consequently, LPHC efficiency actively contributes to sagittal balance during static and dynamic conditions [7]. 

Several authors have analyzed the sagittal balance of the spine and described this component of body posture and a strong relation between lumbopelvic morphology and human ergonomics has been reported [8,9,10,11]. The sagittal alignment of the pelvis represents the base for maintaining postural equilibrium and spinal balance, and the impact of this region on the management of spinal disorders is of significant importance [12]. 

Since it is well documented that global sagittal alignment is associated with the quality of life of people with spine deformities (like scoliosis and thoracic hyperkiphosis), the evaluation of this component is a crucial point in postural assessment in healthy and pathologic conditions [13,14]. The sagittal spinopelvic assessment is commonly executed on lateral radiographs of the spine and pelvis during the standing position using different anatomic and positional parameters [15]. The main investigated pelvic parameters are Pelvic Incidence (angle between the line joining the hip axis and the center of the S1 endplate and a line orthogonal to the S1 endplate); Sacral Slope (angle between the line along the S1 endplate and the reference horizontal line); and Pelvic Tilt (angle between the line joining the hip axis and the center of the S1 endplate and the vertical reference line). The main spinal parameters are Thoracic Kyphosis (angle between the line along the upper T1 endplate and the line along the lower T12 endplate); Lumbar Lordosis (angle between the line along the upper L1 endplate and the line along the lower L5 endplate); the Sagittal Vertical Axis (distance between the C7 plumbline and the posterior–superior S1 corner); and Spino Sacral Angle (angle connecting the center of C7 to the center of the S1 endplate and the line parallel to the superior S1 endplate) [9,12]. These anatomical and positional parameters represent the gold standard in the sagittal balance assessment during medical evaluation.

Postural diseases during childhood and adolescence are strongly related to unbalanced conditions on the sagittal plane, in both structural and non-structural misalignment [12]. Morphologic changes related to skeletal growth and postural control center maturation can alter the spinal alignment and bring unbalanced conditions [13]. Consequently, a proper postural evaluation becomes of primary importance.

In association with radiographic investigation, several non-radiographic methods have been proposed to perform a safe and useful assessment for young patients [16,17]. Inside this scenario, Spinal Mouse^®^ and photogrammetry have been previously investigated and represented quick and non-invasive tools for adolescent idiopathic scoliosis and thoracic hyperkyphosis evaluation [18,19].

Specifically, Spinal Mouse^®^ is a valid and reliable skin-surface mouse that can be used in different body positions (upright standing and forward-bending positions, for example) to analyze spine and pelvis alignment and mobility [20,21]. With regard to the lumbopelvic region, this device has been proposed to investigate the relation between spinal curvature and pelvic tilt in athletes from different sports [22,23]. On the other side, photogrammetry has been widely described and several parameters have been suggested during the postural assessment of healthy and pathologic people. Pelvic Horizontal Alignment (angle formed between the line that links the anterior–superior iliac spine/ASIS and posterior–superior iliac spine/SIPS with a horizontal line); Hip Angle (angle formed between the line that links ASIS and the greater trochanter of the femur and the line that links the joint line of the knee to the greater trochanter); Vertical Body Alignment (angle formed between the line that links the acromion and the lateral malleolus and the vertical line); and Thoracic Kyphosis and Lumbar Lordosis (angle formed from the intersection of the lines that link different spinous processes) represent the main indicators of sagittal balance [24,25,26].

Although the relationship between radiographic and photogrammetric procedures has been previously investigated to quantify specific sagittal parameters, the comparison between Spinal Mouse^®^ and photogrammetry for lumbopelvic assessment is still lacking. Recently, Belli et al. analyzed the relation between these two devices in young people with kyphotic posture, and the main focus was addressed on spinal thoracic parameters [19]. 

Starting from previous work, the present study aims to correlate the lumbopelvic measurements of Spinal Mouse^®^ with some postural parameters obtained with photogrammetry in adolescents with kyphotic posture. 

## 2. Materials and Methods

### 2.1. Design and Participants

This study design is cross-sectional. The sample size needed for controlling type I and II errors has been estimated by an a priori calculation for linear regression with the following features: type I error = 5%, statistical power = 95%, R^2^ = 30%, number of covariances tested = 2. The sample size requested was 40. 

The research was performed at Fisiokinè Medical Center (Scandiano, Reggio Emilia, Italy). Participants were recruited from a sample of patients involved in a physiotherapy approach at the beginning of their treatment. The criteria of selection included a diagnosis of increased thoracic kyphosis (postural or structural hyperkyphosis), age from 10 to 16 years old, no history of musculoskeletal or neurological pain in the last 3 months, and no prior surgical intervention for spine disorders. Since participants were younger than 18 years old, parents’ consent was requested. All participants were informed and gave voluntary consent to participate in the study, and privacy criteria were met. The study was approved by the Bioethics Committee of the University of Bologna and was conducted in accordance with the guidelines of the Declaration of Helsinki; the project identification code was n.2.18 (April 2017). During the recruitment phase, each participant completed the anamnesis investigation, and all medical reports were collected and analyzed to meet the selection criteria. The enrolment phase lasted 12 months, from June 2022 to May 2023.

### 2.2. Measurement Instruments

#### 2.2.1. Spine Analysis

The spinal curves and trunk alignment were evaluated using the SpinalMouse^®^ (IDIAG M360^®^, Fehraltorf, Switzerland) tool. It is a non-invasive computer-assisted medical device that quantifies the curvature and mobility of the spinal column in the frontal and sagittal planes, by gliding it manually along the spine [27,28]. Since this device evidenced excellent intra-rater reliability for the analysis of the sagittal thoracic and lumbar spine and showed high correlation with the radiological Cobb angle on the frontal plane, it has been widely employed in postural disorders with people of different ages [18,19,29,30]. The SM can sample the data every 1.3 mm while the cutaneous mouse is rolled from vertebra C7 to S3, giving a sampling frequency of approximately 150 Hz. Results are wirelessly transferred to a computer, where the IDIAG software displays vertebral positions, joint angles, and spinal alignment. In the current study, a trained specialist with more than five years of experience performed each test. In order to avoid postural effects in the spine due to fatigue or daily stress factors, the evaluation was settled during the morning in a quiet and well-lit environment with a comfortable temperature [18]. After undressing the upper body, the C7–S3 vertebral spinal processes were determined and marked with a dermo graphic pen by the specialist while the patient was standing up in the anatomical position. Measurements were performed in 3 different trunk positions during standing: neutral (upright), maximal flexion, and extension (sagittal plane evaluation). In the neutral position, the participant was asked to maintain a relaxed position, looking and facing horizontally toward the wall, with the feet shoulder-width apart and straight knees and arms by the side. In maximal flexion, the subject was asked to flex the trunk with extended legs as far as possible, aiming to touch the ground with fingertips. In maximal extension, the participant crossed his arms in front of the chest and extended the trunk as far as possible, without extension of the cervical spine. In each position, Spinal-Mouse^®^ was then moved downwards along the spinal criteria points. Participants did not perform a warm-up before the examination. In the present study, some specific angular measures (degrees) on the sagittal plane were extracted and analyzed from all raw data available. The six variables were lumbar spine curvature with fixed upper and lower limits (first and last lumbar vertebra) in standing (FLASt_SM) and during flexion (FLABd_SM), lumbar spine curvature with physiological upper and lower limits (defined by SM software in relation to the thoracic–lumbar inflexion point) in standing (PLASt_SM) and during flexion (PLABd_SM), sacral–hip position in standing (Sacral-hip St_SM) and during flexion (Sacral-hip Bd_SM). Figure 1 shows some of all the possible Spinal-Mouse measurements displayed by IDIAG M360 software and used in the present study, as reported by Belli et al., 2023 [19].

#### 2.2.2. Photogrammetric Postural Analysis

Photogrammetry (PG) has been previously demonstrated to be a reliable method during the assessment of young people with postural disorders [24,31]. Photogrammetric measurements of thoracic kyphosis showed excellent test–retest reliability (ICC = 0.97; SEM = 1.67; MDC = 4.62) in adolescents with hyperkyphosis and evidenced a strong correlation between the values obtained with this technique and radiologic methods [30]. In the current study, two digital photographs on the sagittal plane (standing right side and standing trunk flexion) were recorded using a portable device (Tablet Huawei^®^ Mediapad, Shenzhen, Guangdong, China). The device was set on a tripod, three meters away from the line marking the position of the participant. The height of the tripod was adjusted in order to settle the middle of the objective lens 100 cm above the ground [31]. Evaluations were assessed as by a previous study [19]. Each participant was positioned initially in front of the camera with a postural grid (ATS^®^, Arezzo, Italy) on the back, then the body turned to the left to point the right side perpendicular to the lens, with feet placed in a fixed position over a specific trace (standing right-side position—Appendix A). Successively, participants flexed the trunk and remained in the forward-bending position (standing trunk flexion position—Appendix A). The digital photographs were recorded by a trained specialist during the maintenance of the upright standing and bending positions. The application APECS-AI Posture Evaluation and Correction System^®^ (New Body Technologies SAS, Grenoble, France) was used to evaluate the absolute and relative angles in the sagittal plane [18,25]. The angles investigated were as follows: body vertical inclination (absolute angle between the vertical line and a line connecting the lateral malleolus–tragus of the ear, Tragus-malleolus_PG), trunk flexion (absolute angle between the horizontal line and a line connecting sacral endplate–C7 spinous process, Sacral-C7 Bd_PG), sacral inclination during bending (absolute angle between the horizontal line and a tangent line to sacral dorsum, Sacral tg Bd_PG), hip position during bending (absolute angle between the vertical line and a line connecting the lateral malleolus–greater trochanter, Hip-malleolus Bd_PG), anterior–superior iliac spine (SIAS) –posterior–superior iliac spine (SIPS) angle during standing (absolute angle between the horizontal line and a line connecting SIAS and SIPS, SIPS_SIAS Sd_PG, Figure 2). To better detect previous anatomic landmarks during photographic analysis, an adhesive tape was applied to the skin [31].

### 2.3. Statistical Analysis 

The descriptive statistics have been reported as the mean plus or minus the standard deviation (SD) values for each variable. The variables’ distribution was verified graphically (box plot and histogram with k-density plot) and then confirmed through the Shapiro–Wilk test. If any variable did not meet the distribution assumption, a location-scale transformation (logarithm or exponential (^−1.5^)) was applied. The significance level was settled at ≤0.05 for all inferential statistics. The Pearson product–moment (r) was calculated to measure the degree and the direction of correlation between the Spinal Mouse and photogrammetry variables. To perform the best regression model, the stepwise backward procedure was assessed with a significant level for entry of 0.05 and removal of 0.07. The model’s heteroskedasticity was checked using the Breusch–Pagan/Cook–Weisberg test, with the null hypothesis of homoskedasticity. The multicollinearity was checked using the variance inflation factor (VIF), and a mean value lower than 5 was considered acceptable (moderate correlation) [32]. The Cook’s distance plot was performed to look for the presence of outliers, with a threshold settled at n/4. If one or more outliers affected the model, they were removed, and a new model was performed. Then, the two regression models were compared. Both the adjusted R^2^ and the Akaike information criteria (AIC) were calculated to report the goodness-of-fit and the loss of information for the proposed models. In addition, the Snedecor–Fisher value, the root mean square error (RMSE), the regression coefficients (β), and the intercepts were computed. Finally, the Bland–Altman plot, the pairwise correlation, and the concordance correlation coefficient (CCC) were computed to estimate the degree of agreement between the observed and predicted values [33].

## 3. Results

Table 1 shows the descriptive statistics of the whole sample clustered in male (64.10%) and female (35.90%) participants. Generally, the average angle between the anterior–superior and posterior–superior iliac spine in the standing position was 11.05 ± 5.61°, while both fixed and physiological lumbar angles in the bending position were 36.31 ± 13.16° and 35.87 ± 13.90°, respectively. As regards gender comparison, the sacral tangent angle was the only significant difference among male and female adolescents (F _(1, 37)_ = 8.30, *p* < 0.01, ES = 0.94 [0.26; 1.61]). 

Table 2 shows the pairwise correlation matrix of the Spinal Mouse and photogrammetry variables. The sacral–hip Spinal Mouse evaluation in the bending positions exhibited the highest values of correlation with photogrammetry, especially for the tangent of the sacral angle (r = −0.66) and the angle between the C7 and sacral points (r = 0.49). Although FLABd_SM, PLASt_SM, and FLASt_SM showed a significant correlation with the angle between the C7 and sacral points (r = 0.43) and the angle between the anterior–superior and posterior–superior iliac spine in standing (r = 0.33 and r = 0.38), respectively, the best regression model was fitted by Sacral-hip Bd_SM. 

Table 3 shows the best regression model for the Spinal Mouse sacral–hip angle in the bending position and the two best regressors derived by the stepwise procedure. Due to the presence of three possible outliers, two different models were drawn and compared and model 2 (Table 3) was selected as the final model (adj. R2 = 0.56, ∆AIC = −10.43). The mean VIF for the tangent of the sacral angle and the body mass index was 1.00, while the heteroskedasticity χ^2^ test was 0.16 (*p* = 0.69). 

Appendix A shows the P-P plot of the estimated variable for the distribution assumption. Figure 3 shows the correlation plots of the logarithm of the Spinal Mouse sacral–hip angle and both regressors. Figure 4 shows the Bland–Altman plot in which 5.13% of the observations fell outside of the 95% agreement limit. The concordance correlation coefficient among the observed and estimated values was 0.738 [0.603; 0.873]. 

## 4. Discussion

The present study aimed to find out a model of regression that could relate the analytic lumbopelvic measurements of the Spinal Mouse^®^, with one or more photogrammetric parameters of body posture in adolescents with kyphotic posture. 

The main findings evidence moderate to good correlations both for the standing and bending positions between some measurements of the SM and PG. To the best of our knowledge, this is the second study that compared the SM and PG, following the results previously published on the kyphotic curve [19], and the one that takes into consideration the regression model error due to intra-subject variability (outlier presence). The above-mentioned publication highlighted how photogrammetry could explain 80.4% of Spinal Mouse variability during forward bending. In particular, PG sacrum inclination evidenced a significant correlation with SM trunk inclination and PG hip and trunk angles explained the variability of the SM in this position. Differently, the following study highlights the strong correlation between the sacral–hip angle evaluated in the bending position with the SM and the sacral tangent calculated with PG (*r* = 0.66). These results suggested how the photogrammetric analysis of the pelvic region is deeply connected with spine inclination in this kind of subject and broadened the investigation on sagittal balance. 

Recently, a review by Czubak-Wrzosek et al. [14] analyzed the relationship between sagittal spinopelvic alignment and pediatric pathologies. Concerning the structural kyphotic posture, the occurrence of thoracic or thoracolumbar kyphosis induces a compensatory mechanism on the lumbar and pelvic region, resulting in increased hyperlordosis and decreased pelvic incidence. Thus, the maintenance of sagittal balance requires changes in the pelvic shape during growth and significant modification in postural control occurs before achieving skeletal maturity. The authors conclude that a thorough understanding of sagittal alignment in spine disorders is essential for proper assessment and treatment. Although the normative data about sagittal postural parameters (pelvic tilt, pelvic incidence, sacral slope, sagittal vertical axis, thoracic and lumbar spine angles) have been recently analyzed and proposed using radiologic analysis [11], the use of non-invasive methods must be considered [34,35,36]. In relation to the above-mentioned studies and previous utilization of these tools with similar populations [21,30,37,38,39], the SM and PG were examined in the current study. 

The mean fixed, physiological lumbar spine, and sacral–hip SM angles during upright posture were 30.69°, 33.26°, and 16°, respectively. Differences were reported for male and female values, even if not significant (28.52° and 14.72° for males and 34.57° and 18.29° for females in fixed lumbar and sacral–hip angles, respectively). The male fixed lumbar lordosis angle (FLASt_SM) is similar to the value of 29.4° observed by Rabiezaadeh et al. within a population of 97 healthy boys (mean age 13.8 years old) and to the mean value of 28.37° reported by Lopez Minarro et al. [23] in a sample of young male athletes (32 canoeists, 30 kayakers, 24 tennis players) aged 15–17 years old. A similar connection with this study can also be found for the sacral–hip angle (named “pelvic tilt” by the authors), with a mean value of 13.94° in a standing position. Contrary to our SM results, significant sex-related differences in spine curvature and pelvic tilt were reported in 40 tennis players (24 male and 16 female) aged 13–18 years old [22]. The sample profile and sports features could justify this result [40]. 

Regarding PG, a gender-specific nuance surfaced, unveiling a significant difference in the sacral tangent angle between males and females during the forward-bending position. This underscores the imperative consideration of gender-specific variations in spinal morphology during adolescence, a period marked by dynamic musculoskeletal development and unbalanced conditions [41]. 

On the side of the relationships between the SM and PG, the sacral–hip angle during bending emerged as the protagonist, boasting the highest correlation values. The sacral–hip measured with the SM correlates with the sacral tangent (*r =* −0.66) and with the sacral–C7 angle (*r =* 0.49) measured with PG. The negative value of the first correlation depends on the different reference lines involved in the angular calculation: the SM assigns the value “0” to the vertical line, while PG considers the horizontal line as the starting point. Consequently, the increase in the SM corresponds to a lower PG value for the sacral tangent. Regarding the PG sacral–C7 angle, the horizontal line was assumed as “90°” and the final value was calculated about this point. Thus, a higher bending position corresponds to higher angles for both the SM and PG, eliciting a positive correlation. 

Starting with these results, the main findings of this paper suggest that PG is much more useful for evaluating the bending position than the standing one. It can be considered coherent with the high use of the forward-bending test in spinal assessment [42,43,44]. The forward-bending position is a masterpiece of spinal function assessment; although it is questioned for its clinical validity [45], it is widespread for its quickness and usefulness in showing the subject’s ability to move the spine forward. For this reason, it is very interesting to know that PG measurements are correlated to the SM in the bending position. In the present study, the landmarks for pelvic investigation were defined as reported by Carregaro et al. [46]. Since the validity and reliability of the SM to assess lumbar mobility during trunk flexion has been previously reported [47] and this device is widely employed in bending positions (Sit and Reach and Toe-touch tests), the current investigation can provide further data [48]. 

The other two significant relationships between the SM and PG can be mentioned as interesting. These relationships are moderately present in the standing position; therefore, it is important to discuss them. The correlations are between the SIPS-SIAS angle and the PLASt (*r* = −0.33) and FLASt angles (*r* = −0.38) measured by the SM. The correlation coefficient values are very similar because the angle is very similar as well, and the negative correlation is due to the minus sign used by the SM to detect lordosis. The meaning of this relationship is that the more inclined the pelvis is, the more convex the lordosis is. It is well known according to classic kinesiology; at the same time, it can be considered a novelty from a practical point of view. Measuring the SIPS-SIAS angle with the PG is a very easy and cheap job for posture professionals and can provide interesting information about sagittal balance [26,36]. The main values of these angles are 11.84° for males and 9.64° for females, without highlighting any significant gender difference. Previous research defined this PG parameter as “horizontal alignment of the pelvis” and suggested its analysis to assess anterior and posterior pelvic tilt [35]. The systematic review of Krawcky et al. [26] defined the reference values for several PG parameters and reported them as −12.26°. Positive or negative values identify posterior or anterior pelvic tilt, respectively. Since all participants of the present work showed anterior pelvic tilt, this angle was reported as an absolute value and presented only as a positive number. Consequently, the pelvic morphology and lumbar lordosis obtained are probably related to compensation mechanisms resulting from thoracic kyphosis [11]. The choice of PG parameters in the current study was mainly related to the specificity of APECS-AI applications and the need to perform a quick evaluation, with minimal measurement errors. Thus, the postural alignment was preferred to thoracic and lumbar curvature evaluation [26].

The final finding regards the positive relationship between the sacral–hip angle in the bending position and the body mass index (BMI). Although different results emerged from previous studies, to the best of our knowledge, the correlation between the lumbar curvature and the BMI has been investigated only in the standing position [49,50]. Independently of sexes and ages, the positive relationship suggested a main role for body load in the growing of sagittal spinal curvature in lumbar spine [Smith et al., 2011]. Obesity or overweight in adolescence could negatively affect the neuromuscular and skeletal system development for both the increased load on posture and its effect on the mental state and psychological status [51]. However, many investigations are needed for spine curvature in the bending assessment. 

In conclusion, the main findings of the current paper allow the professional to use PG both in standing and bending positions to assess the spine functionality, accordingly to the analysis on kyphosis previously published [19]. 

These results can be read positively under the light that the SM is highly used in both bending and standing positions [22,23]. A previous paper suggests the validity and reliability of the instrument, allowing us to compare it with PG. It is important to underline that the SM is also used on children, youth, and adolescents because it represents a useful and healthy assessment tool [38,39,40]. At the same time, even PG is commonly used with children, youth, and adolescents to evaluate posture alignment [30,35]. The novelty of the current paper lies in the discovery of significant relationships between two instruments that are commonly used separately. It is not the focus of this paper to assess which instrument could be more suitable for these subjects. Contrarily, the aim is to give evidence about the possibility of assessing similar postural spine behaviors with different tools, to help professionals in their everyday job. The current study can be considered in the same theoretical and cultural frame as our previous paper published on the thoracic kyphosis assessment comparing the SM and PG. With the current study, the spine assessment was integrated by adding evidence on the lumbar lordosis. 

### Limitations

However, in the quest for scientific rigor, it is fundamental to acknowledge the study’s limitations. The cross-sectional nature inherently curtails the ability to draw causal inferences. Moreover, the relatively modest sample size necessitates circumspection in extending findings to broader populations. Indeed, the sample only contains participants with augmented thoracic kyphosis, and it could be interesting to replicate the study even on other clusters of typical postural adolescent imbalances. Future research should chart a course toward longitudinal designs and more expansive cohorts to fortify the generalizability of these pivotal findings. In the end, a methodological issue should be underlined as a limitation. The use of PG and the SM can be affected by the operator’s experience. In the current study, the expertise of the operator was high; therefore, it should be interesting to replicate the measurements between different operators with different experience levels to understand how the expertise could or not affect the relationships between the SM and PG.

## 5. Conclusions

Photogrammetry is recognized as a straightforward, cost-effective, and rapid method for the first-level assessment of spinal posture. It appears to accurately reflect actual spinal dynamics in both upright and bending postures, particularly for examining lumbar and sacral angles in adolescents with pronounced thoracic kyphosis. Notably, the forward-bending posture provides the most reliable regression model for assessing lumbopelvic alignment on the sagittal plane. Therefore, kinesiologists and other specialists engaged in postural evaluation are recommended to adopt photogrammetry as a primary tool for assessing the sagittal spinal configuration in adolescents.

## Figures and Tables

**Figure 1 healthcare-12-00738-f001:**
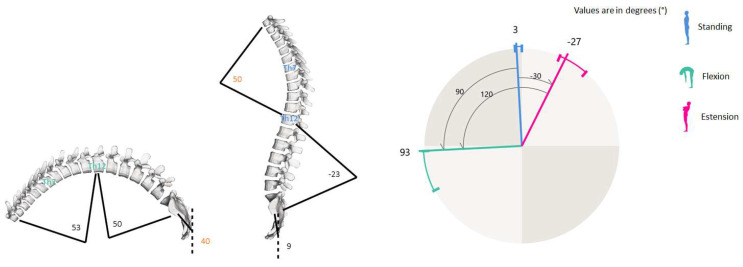
Spinal Mouse evaluations in bending (**left**) and standing (**right**) positions. The purple angle range represents the hypothetical assessment of maximal spine extension. Note: This figure is adapted from Belli et al., 2023 [19].

**Figure 2 healthcare-12-00738-f002:**
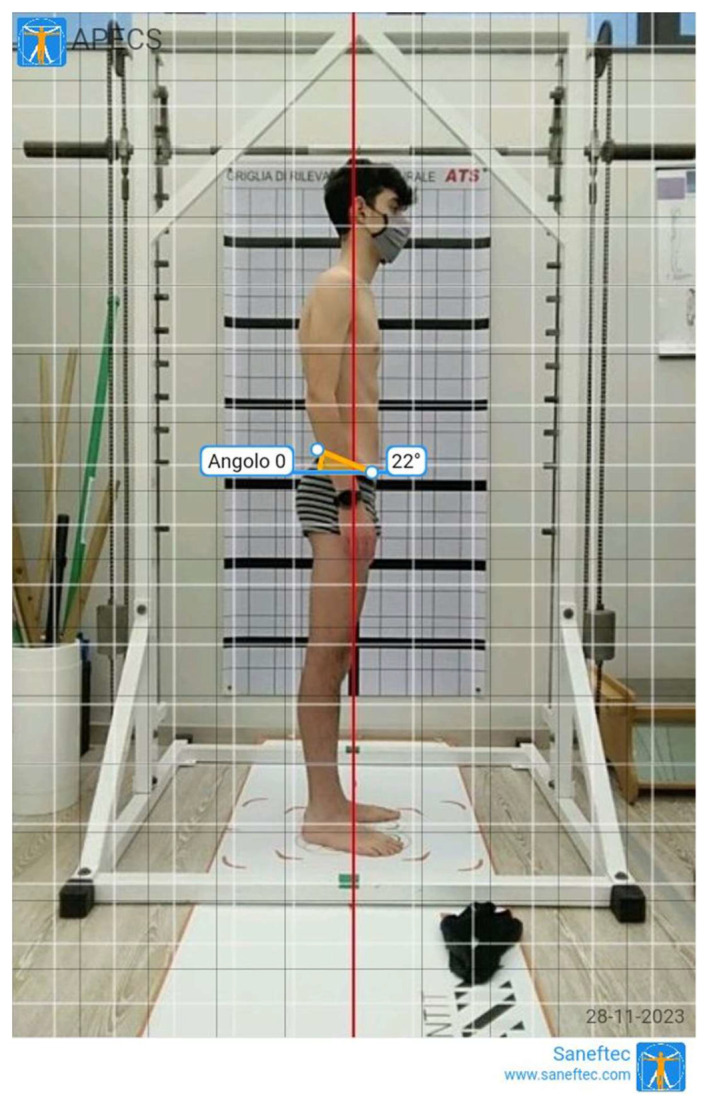
SIPS and SIAS evaluation assessed with photogrammetry.

**Figure 3 healthcare-12-00738-f003:**
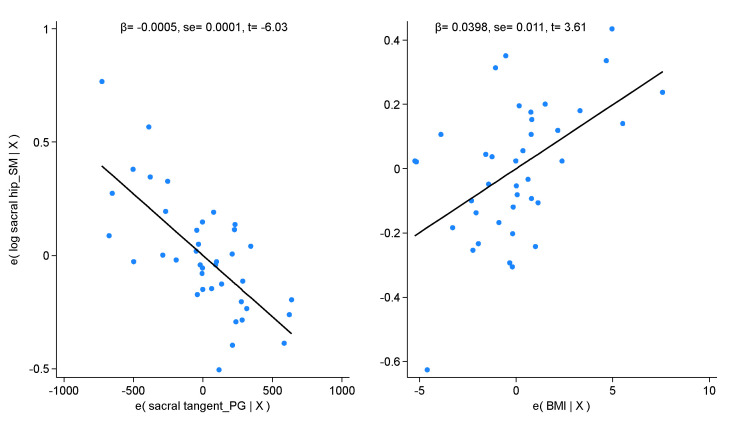
Average plots of each regressor and the predicted model.

**Figure 4 healthcare-12-00738-f004:**
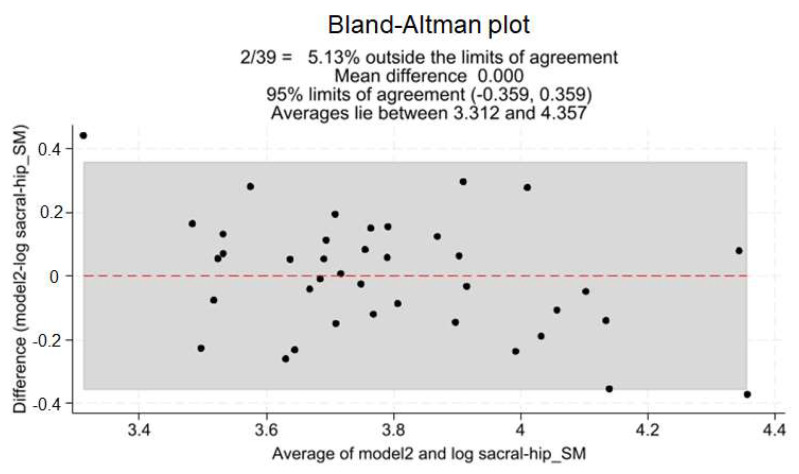
Bland–Altman plot for regression model agreement.

**Table 1 healthcare-12-00738-t001:** Descriptive characteristics of male and female participants.

	Gender			
	Male (n = 25)		Female (n = 14)	
	Mean	SD	Mean	SD
Age [year]	13.04	2.11	13.57	1.22
Stature [cm]	159.52	14.67	158.79	6.48
Body mass [kg]	49.20	12.96	44.79	9.13
BMI [kg/m^2^]	19.02	2.82	17.66	2.54
SIPS_SIAS St_PG [°]	11.84	5.61	9.64	5.53
Tragus-malleolus_PG [°]	2.70	1.37	2.81	1.03
Sacral-C7 Bd_PG [°]	89.32	10.83	96.21	14.70
Sacral tg Bd_PG [°]	41.48	8.66	30.93	14.29
Hip-malleolus Bd_PG [°]	8.84	2.87	7.57	2.24
FLASt_SM [°]	−28.52	10.88	−34.57	11.88
PLASt_SM [°]	−30.92	11.10	−37.43	9.31
FLABd_SM [°]	38.68	13.05	32.07	12.72
PLABd_SM [°]	38.28	12.95	31.57	14.97
Sacral-hip St_SM [°]	14.72	8.37	18.29	6.22
Sacral-hip Bd_SM [°]	44.24	10.22	49.57	19.52

Note: SD, standard deviation; BMI, body mass index; St, standing position; Bd, bending position; PG, photogrammetry; SM, Spinal Mouse; SIPS, posterior–superior iliac spine; SIAS, anterior–superior iliac spine; FLASt, fixed lumbar angle; PLASt, physiological lumbar angle; FLABd, fixed lumbar angle; PLABd, physiological lumbar angle.

**Table 2 healthcare-12-00738-t002:** Pearson’s product–moment correlation matrix of the Spinal Mouse and photogrammetry parameters.

	I	II	III	IV	V	VI	VII	VIII	IX	X	XI
(I) Sacral-hip Bd_SM	1										
(II) Sacral-hip St_SM	0.0905	1									
(III) PLABd_SM	−0.3613 *	−0.1528	1								
(IV) FLABd_SM	−0.3555 *	−0.3965 *	0.7809 *	1							
(V) PLASt_SM	0.0202	−0.8252 *	0.2309	0.3649 *	1						
(VI) FLASt_SM	0.0253	−0.8638 *	0.2267	0.4560 *	0.9439 *	1					
(VII) Sacral tg Bd_PG	−0.6595 *	−0.0878	0.0544	−0.0182	0.0793	−0.0061	1				
(VIII) SIPS_SIAS St_PG	0.0076	0.2713	−0.0438	−0.126	−0.3283 *	−0.3791 *	0.022	1			
(IX) Tragus-malleolus_PG	−0.0095	0.3094	−0.1295	−0.1621	−0.3056	−0.2643	−0.0802	0.1711	1		
(X) Sacral-C7 Bd_PG	0.4901 *	−0.0823	0.2982	0.4325 *	0.0656	0.1889	−0.8045 *	−0.0913	−0.0204	1	
(XI) Hip-malleolus Bd_PG	−0.0066	−0.1252	−0.2517	−0.2366	0.2021	0.154	0.1716	0.2331	−0.1322	−0.4278 *	1

Note: Bd, bending position; SM, Spinal Mouse; St, standing position; PG, photogrammetry; PLABd, physiological lumbar angle; FLABd, fixed lumbar angle; PLASt, physiological lumbar angle; FLASt, fixed lumbar angle; SIPS, posterior–superior iliac spine; SIAS, anterior–superior iliac spine; *, *p* < 0.05.

**Table 3 healthcare-12-00738-t003:** Regression models for the Spinal Mouse sacral–hip angle in the bending position.

Variable	Model1	Model2
sacral tangent PG	−0.001 ***	−0.001 ***
BMI	0.032 **	0.040 ***
Intercept	3.572 ***	3.454 ***
N	36	39
R^2^	0.583	0.585
Adj. R^2^	0.557	0.562
RMSE	0.159	0.188
AIC	27.172	16.741

Note: N, sample size; R^2^, model goodness-of-fit; RMSE, root mean square error; AIC, Akaike information criterion; **, *p* < 0.01; ***, *p* < 0.001.

## Data Availability

Data are available on request to the first author.

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
