# Peer review of "Relation between Photogrammetry and Spinal Mouse for Lumbopelvic Assessment in Adolescents with Thoracic Kyphosis"

_healthcare, 2024, doi:10.3390/healthcare12070738_

Round 1

Reviewer 1 Report

Comments and Suggestions for Authors

This is a nice study evaluating posture using spinal mouse and photogrammetry. The authors measured both ways and compared the results from both measurements.

The discussion and conclusions have to be focused towards the objective of this study. Right now the discussion and conclusions are not specific to the results and finding.

Comments on the Quality of English Language

English is good and a spell and grammer check should be done.

Author Response

Dear Reviewer 1,

We would like to thank you for the time allowed to this review process. As a result, we are submitting the revised version for a possible publication in this respectable Journal. Below, you can find our responses; each comment is followed by its respective reply. We made changes in the manuscript in order to address suggestions and make it clearer for the readers. Our responses in the manuscript appear using the track change instrument. We very much appreciate your comments on the document, which have helped us to improve its quality.

All authors have made sufficient contributions, responded to your comments and have approved the submitted manuscript.

Best regards,

The Authors

Legend:

R1(Reviewer 1)

A (Authors)

1) R1:

This is a nice study evaluating posture using spinal mouse and photogrammetry. The authors measured both ways and compared the results from both measurements.

A: We thank the Reviewer 1 for appreciating our work and we provide some modifications in the following lines to improve the paper as you suggest.

2) R1:

The discussion and conclusions have to be focused towards the objective of this study. Right now the discussion and conclusions are not specific to the results and finding. 

A: Dear Reviewer, from our point of view the results have been described and compared with literature in the discussion paragraph. In particular, we mainly focused on the descriptive data and correlation values presented in Tables 1 and 2, respectively. This choice is related to the greater comparison with previous research and the practical understanding of results. Table 3 discusses the importance of bending position analysis during postural assessment. However, we improved the discussion section to better explain and link recent results with previous investigations. Also, the conclusion paragraph has been improved

Reviewer 2 Report

Comments and Suggestions for Authors

Formal examination:

The topic of the publication is extremely interesting and is devoted to a comparative analysis of the effectiveness of non-X-ray methods for assessing the sagittal profile of the spine in adolescents. This also seems important for remote assessment of spinal posture and screening for deformities.

Design: single-center cohort comparative analysis.

The goal is clearly stated. I would also like to note the clear structure of the work.

Content expertise:

1. It is necessary to clarify by what method criteria 4 - 14 in Table 1 were obtained?

General assessment:

Overall, the work is extremely interesting to read and makes a good impression. Minimal correction recommended.

I wish the authors success.

Author Response

Dear Reviewer 2,

We would like to thank you for the time allowed to this review process. As a result, we are submitting the revised version for a possible publication in this respectable Journal. Below, you can find our responses; each comment is followed by its respective reply. We made changes in the manuscript in order to address suggestions and make it clearer for the readers. Our responses in the manuscript appear using the track change instrument. We very much appreciate your comments on the document, which have helped us to improve its quality.

All authors have made sufficient contributions, responded to your comments and have approved the submitted manuscript.

Best regards,

The Authors

Legend:

R2 (Reviewer 2)

A (Authors)

1) R2:

Formal examination:

The topic of the publication is extremely interesting and is devoted to a comparative analysis of the effectiveness of non-X-ray methods for assessing the sagittal profile of the spine in adolescents. This also seems important for remote assessment of spinal posture and screening for deformities.

Design: single-center cohort comparative analysis.

The goal is clearly stated. I would also like to note the clear structure of the work.

A: We thank the Reviewer 2 in appreciating our work. We will provide proper answers in the following lines. We think that the language corrections suggested by the Reviewer can improve the paper quality.

2) R2:

Content expertise:

  1. It is necessary to clarify by what method criteria 4 - 14 in Table 1 were obtained?

A: Dear reviewer, the specific values are described in Spinal Analysis and Photogrammetric Analysis paragraphs. More detailed Spinal Mouse description has been added in 2.2.1 section. 

3) R2:

General assessment:

Overall, the work is extremely interesting to read and makes a good impression. Minimal correction recommended.

I wish the authors success. 

A: We are grateful for the appreciation of Reviewer 2.

Reviewer 3 Report

Comments and Suggestions for Authors

Please refrain from one- two-sentence paragraphs. 

Please recheck whether the keywords exist in MeSH browser. 

All used software should be given along with manufacturer and country of origin. 

Please add date and number of ethical approval. 

How was the sample size arrived at? 

Table 2: Do the values represent r values? 

Did any of the patients had coronal malalignment? 

Does each author meet the 1st criterion of the ICMJE 4 criteria? 

Did any of the patients have LSTV deformity, which would have directly interfere with the results of the spinopelvic alignment. Please, at least, provide an insight to this point in the discussion section. Following items might help: PMID: 38106303   PMID: 37497739  PMID: 38030157

Author Response

Dear Reviewer 3,

We would like to thank you for the time allowed for this review process. As a result, we are submitting the revised version for a possible publication in this respectable Journal. Below, you can find our responses; each comment is followed by its respective reply. We made changes in the manuscript in order to address suggestions and make it clearer for the readers. Our responses in the manuscript appear using the track change instrument. We very much appreciate your comments on the document, which have helped us to improve its quality.

All authors have made sufficient contributions, responded to your comments and have approved the submitted manuscript.

Best regards,

The Authors

Legend:

A (Authors)

1) Please refrain from one- two-sentence paragraphs.

A: Thanks for the comment, we made some modifications to the text.

2) Please recheck whether the keywords exist in MeSH browser.

A: Keywords have been modified with specific ones in MeSH

3) All used software should be given along with manufacturer and country of origin. 

A: The software details are already present in the specific paragraph (Spinal Mouse and APEC tools).

4) Please add date and number of ethical approval. 

A: Dear Author, we added the date as you requested. The number has been previously presented in the manuscript. 

5) How was the sample size arrived at? 

A: Thanks for your comment. We added specific information in lines103-106.

6) Table 2: Do the values represent r values? 

A: Thank you for your comment. Of course, it is explained in lines 200-202. However, we have also inserted it in Table 2.

7) Did any of the patients had coronal malalignment? 

A: Thanks for the correct and interesting question. All patients had a diagnosis of increased thoracic kyphosis, and only a few of them had small coronal misalignment (evaluated with TRACE assessment from clinicians). The TRACE score and angle of trunk rotation during bending were under the limits of scoliosis diagnosis.

8) Does each author meet the 1st criterion of the ICMJE 4 criteria?

A: Yes, all the Authors contributed substantially to the paper and the Author's contribution is well described in the final part of the paper.

9) Did any of the patients have LSTV deformity, which would have directly interfere with the results of the spinopelvic alignment. Please, at least, provide an insight to this point in the discussion section. Following items might help: PMID: 38106303   PMID: 37497739  PMID: 38030157

A: Thanks for the detailed analysis. No patients had this Syndrome or other lumbo-pelvic deformities. In addition, the pain condition represented an exclusion criterion.

Reviewer 4 Report

Comments and Suggestions for Authors

I read the article with interest. 

There are few paragraph rearrangement is needed.

The conclusion must rewrite with confident way of presentation with the results.

Also, figures of spinal mouse application will help a lot for readers to understand spinal mouse usage.

Thankl you.

Author Response

Dear Reviewer 4,

We would like to thank you for the time allowed for this review process. As a result, we are submitting the revised version for a possible publication in this respectable Journal. Below, you can find our responses; each comment is followed by its respective reply. We made changes in the manuscript in order to address suggestions and make it clearer for the readers. Our responses in the manuscript appear using the track change instrument. We very much appreciate your comments on the document, which have helped us to improve its quality.

All authors have made sufficient contributions, responded to your comments and have approved the submitted manuscript.

Best regards,

The Authors

Legend:

A (Authors)

1) I read the article with interest. There are few paragraph rearrangement is needed.

A: Thanks for the comment. We modified some parts of the text.

2) The conclusion must rewrite with confident way of presentation with the results.

A: The conclusion has been modified.

3) Also, figures of spinal mouse application will help a lot for readers to understand spinal mouse usage. 

A: Dear reviewer, thank you for your suggestion. We added "Figure 1" as you requested. 

4) source PDF. Line 111 “vol-untary” spell check 

A: Done, line 114.

5) source PDF. Lines 114-115 connect the 2 paragraphs  

A: Done.

6) source PDF. Line 297-303 since those two sentences are about gender, connecting those two sentences in one paragraph is recommended. 

A: Done.

7) (source PDF). Line 384. The conclusion must begin with the findings and recommendation from the results and discussion. The current conclusion doesn't sound confident with the spinal mouse use. Also, "in conclusion" is repeated in the conclusion. Revision is necessary. 

A: Conclusion has been modified

Round 2

Reviewer 4 Report

Comments and Suggestions for Authors

The article has been well re-written.

Author Response

Dear Reviewer,

thank you for your comment!